# Control Strategy of Electric Heating Loads for Reducing Power Shortage in Power Grid

**Siyuan Xue [1], Yanbo Che [1,\*], Wei He [2], Yuancheng Zhao [1] and Ruiping Zhang [3]**

[1]  Key Laboratory of Smart Grid of Ministry of Education, Tianjin University, Tianjin 300072, China; 1029348750@163.com (S.X.); cheng528zyc@163.com (Y.Z.)
[2]  State Grid Jiangxi Electric Power Research Institute, Nanchang 330096, China; lanlyhw@163.com
[3]  College of Automation and Electrical Engineering, Lanzhou Jiaotong University, Gansu 730070, China; zhrp74@mail.lzjtu.cn
\*  Correspondence: lab538@163.com; Tel.: +86-166-026-10538

**Abstract:** With the development of demand response technology, it is possible to reduce power shortages caused by loads participating in power grid dispatching. Based on the equivalent thermal parameter model, and taking full account of the virtual energy storage characteristics presented during electro-thermal conversion, a virtual energy storage model suitable for electric heating loads with different electrical and thermal parameters is proposed in this paper. To avoid communication congestion and simplify calculations, the model is processed by discretization and linearization. To simplify the model, a control strategy for electric heating load, based on the virtual state ofcharge priority list, is proposed. This paper simulates and analyzes a control example, explores the relevant theoretical basis affecting the control effect, and puts forward an optimization scheme for the control strategy. The simulation example proved that the proposed method in this paper can reduce power storage in the grid over a long period of time and can realize a power response in the grid.

**Keywords:** power shortage; electric heating load; electric water heater; demand response; virtual energy storage (VES), virtual state of charge (VSOC)

## 1. Introduction

When a power shortage occurs in the generation side of the power system, the system frequency will be reduced, resulting in a series of power quality problems. Demand response [1,2] can solve the mismatch problem between supply and demand at a relatively lower cost, which is of great significance to absorb new energy sources, reduce power shortages, and reduce environmental pollution [3–5]. The popularization of advanced metering infrastructure [6] makes it possible to use the control strategy of thermostatically controlled load (TCL) and scale the application of trunked dispatching.

As an important part of the demand response, TCLs realize virtual power storage through indirect heat energy storage. The main principle of electric heating load is Joule's law, that is, the chamber temperature is maintained within a certain temperature range through the on-off state of resistance wire heating. On the premise of guaranteeing users' basic comfort [7], electrical heating equipment, such as electric water heaters [8,9], air conditioners, refrigerators, can be equivalent to virtual power storage devices, so electric heating loads can be regarded as good TCLs. An electrical heating device forced to shut down for several minutes can reduce power consumption by a small amount without dropping the temperature below the comfortable temperature range. When a large amount of electric heating loads are forced to shut down in order, they can release a large amount of electric power. The indirect energy storage capability of electric heating loads can reduce peak load and improve the reliability of power grid operation.

Several studies on the virtual energy storage (VES) model of electric heating loads have been undertaken. Reference [10] analyzed the mechanism of aggregated load oscillations caused by the traditional temperature adjusting method, on the basis of the equivalent thermal parameter (ETP) model, and modeled household electric heating load. However, the rationality for model linearization has not been quantitatively analyzed. Reference [11] presented a temperature state priority list method to suppress power flow and decreased power shortage demand for a given micro-grid; however, the modeling of the working state of the constant temperature state was not accurate enough. Reference [12] presented a load model suitable for terminal voltage control of electric water heaters, which could reduce the peak load of the power grid while ensuring the comfort of users, but the modeling process was not described in detail. In [13], using the characteristics of household electric heating loads such as electric water heaters, a high-precision model reflecting different working conditions of electric heating loads was proposed. However, due to the limitation of computational complexity, this model was suitable for small-scale regulation and control only, instead of for large power grid-trunked dispatching.

ETP modelling, as the theoretical basis of control, is widely used [14]. However, previous studies lack an analysis of power parameters, the complete VES index system, and the deep mining of the coupling relationship between variables, and do not accurately reflect the actual electro-thermal conversion relationship.

In terms of the control algorithm, reference [15] started from the macro-layer of the grid side and the micro-layer of the load aggregator, and presented a bi-level optimal dispatch and control model for air-conditioning loads based on direct load control. But the running states of loads before control right transferring need to be uniformly distributed. Reference [10] proposed a new temperature-adjusting method on the basis of the ETP model, to avoid load oscillations caused by the traditional temperature regulation method, but the parameters of the devices participating in the demand side response needed to be the same. Reference [16] proposed a demand-side decentralized control strategy with variable participation to provide directional control of the start-up and shutdown of TCLs, so as to improve the frequency regulation capability of isolated microgrid systems in collaboration with energy storage systems. But, the operation of the units in the cluster control was not analyzed. Reference [17] developed a weighting coefficient-queuing algorithm based on a modified coloredpower algorithm state-queuing model, which can be used to directly control the TCLs of electric heating equipment.

The daily load peak of a power grid generally lasts for several hours, thus the transfer of control rights can last from a few minutes to hours. Previous studies have paid less attention to the analysis of the control effect in the case of long-term (several hours) transfer of control rights and theoretical analysis of factors affecting the control effect.

In this paper, a load model and a trunked dispatching strategy for electric heating loads are analyzed deeply to solve the problem of electric heating loads with different parameters and demands participating in demand response at the same time. Based on a simplified first-order ETP model, a VES model, which can reflect the electro-thermal exchange, is proposed where the potential of demand response can be fully exploited. Based on this model, the trunked dispatching strategy based on virtual state of charge (VSOC) priority list, is proposed. The control effect under the condition of long-term control right transferring is analyzed, and the control strategy is optimized according to the analysis results. The validity and advancement of the optimized control strategy based on the VSOC priority list are proved by design and simulation examples.

## 2. Virtual Energy Storage (VES) Model of Electric Heating Load

### 2.1. Concepts of VES

When power consumption is increased (or reduced) by controlling the difference between the working states of the equipment before and after the control right transferring of the equipment, and this power is stored in other forms, the equipment can be equivalent to VES. When the electric power consumption of the equipment after control right transferring is greater than that before

transferring, it can be considered to be virtual energy storage charging, whereas, when the electric power consumption of the equipment after control right transferring is less than that before transferring, it can be regarded as VES discharging. The control system of VES can offset the shortage of energy storage by guiding and intervening in energy demand, and can achieve the effect of reducing the energy storage capacity and cost.

VES is described using four indicators—charging/discharging power, switch state, charge/discharge time, and VSOC—which are defined as follows:

(1)  Charging/discharging power: charging power is the difference in power consumption of the equipment after control right transferring minus that before transferring. A VES is in the charging state when the charging power is positive, and in the discharging state when the charging power is negative. The value of the discharging power is the opposite of the charging power;

(2)  Switch state: refers to the switching state of electric heating equipment;

(3)  Charge time: the length of time of the charging state; and discharge time: the length of time of the discharging state; and

(4)  VSOC: The United States Advanced Battery Consortium defines state of charge (SOC) as the ratio of the residual electricity to the rated capacity under the same conditions at a certain discharge rate. Similarly, virtual state of charge (VSOC) is defined as the ratio of the residual energy to the rated capacity under the same conditions at a certain charging and discharging power, which represents the responsiveness of VES at a given stage.

## 2.2. Equivalent Thermal Parameter (ETP) Model of Electric Heating Load

The main idea of the ETP modeling method is to equivalent the internal and external environment parameters of the room (chamber) and the refrigerating (heating) capacity of electric energy conversion, to circuit components, such as resistors, capacitors, and power supplies, then use circuit knowledge to analyze the relationship between temperature and energy conversion.

Considering the process of heat exchange between the medium and the mass in the room (chamber), and the exterior environment, the differential equation of the second-order ETP model is:

$$\dot{x} = Ax + Bu \qquad \dot{y} = Cy + Du$$

$$A = \begin{bmatrix} -(\frac{1}{R_2 C_a} + \frac{1}{R_1 C_a}) & \frac{1}{R_2 C_a} \\ \frac{1}{R_2 C_m} & -\frac{1}{R_2 C_m} \end{bmatrix} \qquad B = \begin{bmatrix} \frac{T_{out}}{R_1 C_a} + \frac{\eta P_{ele}}{C_a} \\ 0 \end{bmatrix}$$

$$\dot{x} = \begin{bmatrix} \dot{T}_{in\_g} \\ \dot{T}_{in\_m} \end{bmatrix} \quad \dot{y} = \begin{bmatrix} T_{in\_g} \\ T_{in\_m} \end{bmatrix} \quad u = 1 \quad C = \begin{bmatrix} 1 & 0 \\ 0 & 1 \end{bmatrix} \quad D = 0$$

(1)

The thermal energy storage process is described by heat capacity and heat transfer resistance. In Equation (1), $P_{ele}$ represents electric power, $\eta$ represents refrigeration or heating efficiency, and $\eta P_{ele}$ is refrigeration or heating power (kW). $T_{in\_g}$ represents the temperature of the medium in the room (chamber)(°C), $T_{out}$ represents the ambient temperature (°C), $T_{in\_m}$ represents the temperature of the mass in the room (chamber) (°C), $C_e$ represents the heat capacity of the medium (J/°C), $C_m$ represents the heat capacity of the mass (J/°C), $R_1$ represents the heat transfer resistance of energy between the interior and the exterior environment of the room (chamber) (°C/W), and $R_2$ represents the heat transfer resistance of energy between the medium and the mass in the room (chamber) (°C/W).

The widely used second-order ETP model [18] is shown in Figure 1.

When the temperature change is relatively smooth, there is no obvious difference between the medium and the mass temperatures. In order to improve the practicability of the model, assuming $T_{in\_g} = T_{in\_m} = T_{in}$, the second-order ETP model can be reduced to the first-order ETP model:

$$\frac{T_{in} - T_{out}}{R_1} + C_e \frac{dT_{in}}{dt} = \eta P_{ele}$$

(2)

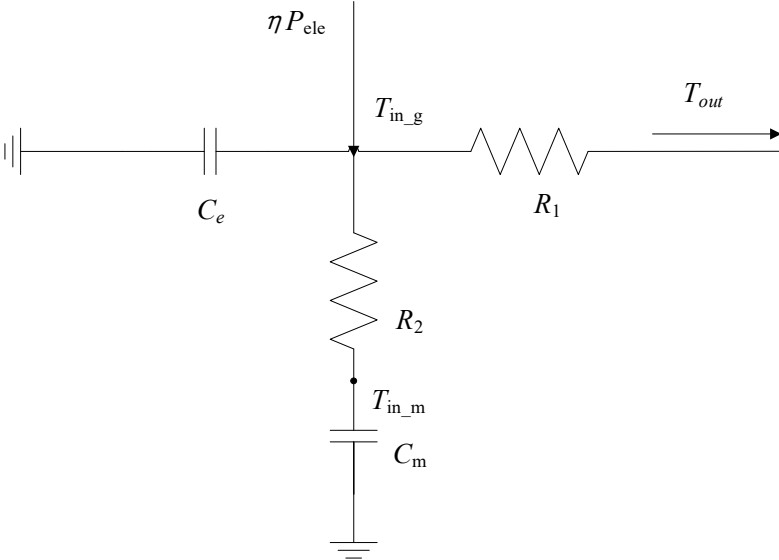

**Figure 1.** Second-order equivalent thermal parameter (ETP) model.

*2.3. Thermal Parameters Part of VES Model for an Electric Heating Unit*

On the basis of the first-order ETP model, a partial model of VES thermal parameters was established as shown in Figure 2.

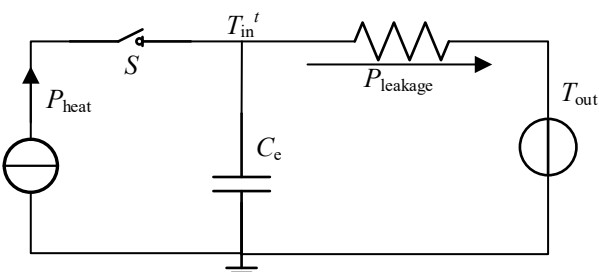

**Figure 2.** Thermal parameters part of the virtual energy storage (VES) model.

$T_{\text{in}}^t$ represents the temperature (°C) of the room (chamber) at time $t$, $S$ represents the switch state, with 0 representing disconnection (the device is closed) and 1 as closure (the device is open). $P_{\text{heat}} = \eta P_{\text{ele}}$ represents the VES power supply, whose specific form depends on the electric part. $P_{\text{leakage}} = (T_{\text{in}}^t - T_{\text{out}})/R_1$ represents the current leakage in the model, mainly represented by the energy loss caused by the temperature gap between the interior and exterior environment.

The electric power before control right transferring is set as the base power, $P_{\text{base}}$. Electric heating equipment that is not in use does not have any discharge capability, and $P_{\text{base}}$ is 0. Charging/discharging power is closely related to the switching state, which directly reflects the real-time reserve energy resource requirements. If $P_{\text{ele}}$ increases when the electric heating equipment participates in the demand response, and it can be considered that the VES is in charging state, and vice versa for the discharge state. Charging and discharging power can be expressed as:

$$P_{\text{disc}} = -S(t)P_{\text{heat(t)}} + P_{\text{base}}$$
$$P_{\text{char}} = S(t)P_{\text{heat(t)}} - P_{\text{base}}$$

(3)

where, the subscript *char* represents charging power and *disc* represents discharging power. $S(t)$ represents the switching state at time $t$. It can be seen that the charging/discharging capacity of VES

comes from the change of state rather than the duration of the state of batteries. Charge/discharge time determines the sustainable response ability of demand resources. If $T_{in}(t_0) = C$, the solution of (2) is:

$$T_{in}(t) = T_{out}(t) + S\eta R_1 P_{ele}(t) - (T_{out}(t) + S\eta R_1 P_{ele}(t) - C)e^{-\frac{t}{R_1 C_e}} \tag{4}$$

and charge/discharge time is:

$$t_{on/off} = R_1 C_e \ln\left(\frac{C - T_{out}(t) - S\eta R_1 P_{ele}(t)}{T_{in}(t) - T_{out}(t) - S\eta R_1 P_{ele}(t)}\right) \tag{5}$$

Using the relationship between temperature and power in ETP model, the maximum capacitance of VES is:

$$Q_{capacity} = C_e(T_{max} - T_{min}) \tag{6}$$

where $T_{max}$, $T_{min}$ are protocol maximum temperature and protocol minimum temperature after the control right is transferred, respectively. Based on the principle of energy conservation, the charge capacity at time $t$ is

$$Q(t) = Q(t_0) + \int_{t_0}^{t} (\eta P_{ele}(\xi) - P_{leakage})d\xi \tag{7}$$

The ratio of residual energy to rated capacity under the same conditions can be expressed by:

$$VSOC = \frac{Q(t)}{Q_{capacity}} \tag{8}$$

Figure 3 shows the charge/discharge curves of two VES systems. Number 1 is denoted by solid lines and number 2 by dotted lines. They participate in the response at $t_1$, and their discharging powers are $P_{char\_1}$ and $P_{char\_2}$ respectively. Since it is convenient to control the equivalent VES of electric heating load, the power climbing state during the response process is neglected. $t_{on\_1}$ and $t_{on\_2}$ represent the discharge time, generally less than the maximum discharge time and limited by the VSOC state of virtual energy storage. For different VESs, their charging/discharging power and charge/discharge time are quite different, but their change modes are the same. The charging state is similar to the discharging state, so it is not necessary to elaborate.

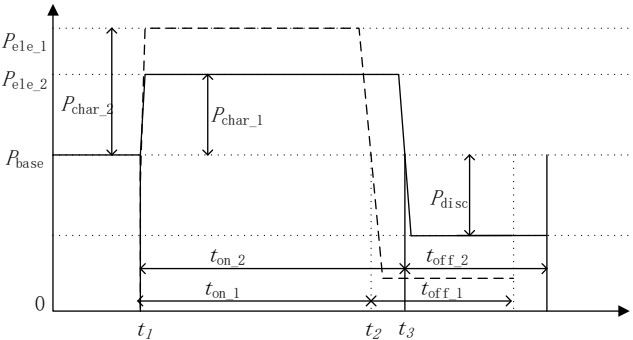

**Figure 3.** Charge-discharge curves of two VES systems.

## 2.4. Electrical Parameters Part of VES Model for An Electric Water Heater

Here we analyze the electrical parameters of a VES model of an electric heating load, and take an electric water heater as an example. An electric water heater keeps the chamber temperature within a specific temperature range by switching the on-off mode of resistance wire, and acts soon after receiving a control signal. As there is no delay when it starts up or turned off, its working state is single. There is no obvious power shock in the water heater. Under the working mode of rated voltage

and rated current, temperature control is realized by switching on/off the devices. The relationship between heating power and electric power is as follows:

$$P_{\text{heatl}} = \eta P_{\text{rated}} \cdot S(t)$$

$$S(t) = \begin{cases} S(t-1) & T_{\min} < T_{\text{in}}(t) < T_{\max} \\ 0 & T_{\max} \leq T_{\text{in}}(t) \\ 1 & T_{\min} \geq T_{\text{in}}(t) \end{cases} \tag{9}$$

where $S(t)$ represents the switching state of the electric water heater at time $t$, $S(t-1)$ represents the switching state at the last time step, 1 for running and 0 for stop; $P_{\text{heatl}}$ represents heating power (kW); $P_{\text{rated}}$ is the rated electric power of electric water heater (kW); $T_{\text{in}}(t)$ represents the chamber temperature at time $t$ (°C).

As shown in Figure 4, the VES model mainly includes the electric power curve, the electrical part and the thermal part. The arrowed solid line indicates the energy flow, and the arrowed dotted line indicates the signal flow.

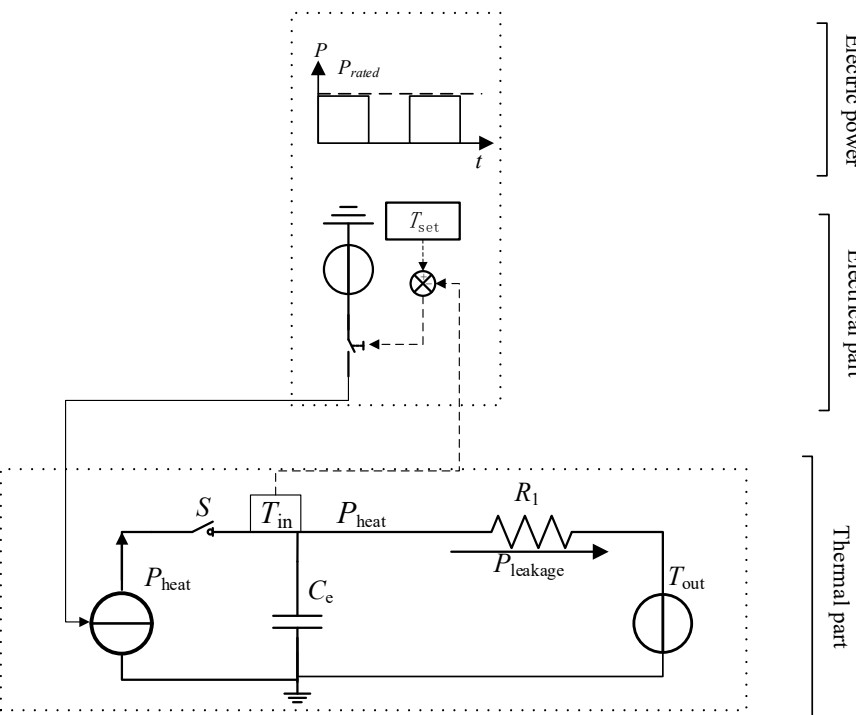

**Figure 4.** VES model of electric water heater.

## 3. Model Preprocessing

In advanced metering infrastructures (AMI), only discrete data are transmitted, and communication time intervals exist. It is necessary to discrete the original VES model. Furthermore, model linearization is also needed to simplify the calculation and reduce communications traffic.

### 3.1. Discretization

Assume that the time step is $\Delta t$, (3) can be expressed as:

$$T_{\text{in}}{}^{t+1} = T_{\text{out}}{}^{t+1} + S\eta R_1 P_{\text{ele}} - (T_{\text{out}}{}^t + S\eta R_1 P_{\text{ele}} - T_{\text{in}}{}^t)e^{-\frac{\Delta t}{R_1 C_e}} \tag{10}$$

where $T_{\text{in}}{}^t$ and $T_{\text{in}}{}^{t+1}$ are the internal temperature of the room(chamber) at time $t$ and $t+1$, respectively; $T_{\text{out}}{}^t$ and $T_{\text{out}}{}^{t+1}$ are the external ambient temperature at time $t$ and $t+1$, respectively.

By connecting the discrete dots calculated by Equation (10) into a smooth curve, the relationship between time and temperature can be accurately described. Figure 5 shows the change of electric power and chamber temperature over time, where $T_{set}$ is the setting temperature.

In $0 \sim t_1$, the water heater is heated from the initial temperature to the protocol maximum temperature. In $t_1 \sim t_2$, the water heater stops heating until the temperature drops to the protocol minimum temperature. The dot dash expresses the temperature drop curve after the unit is shut down.

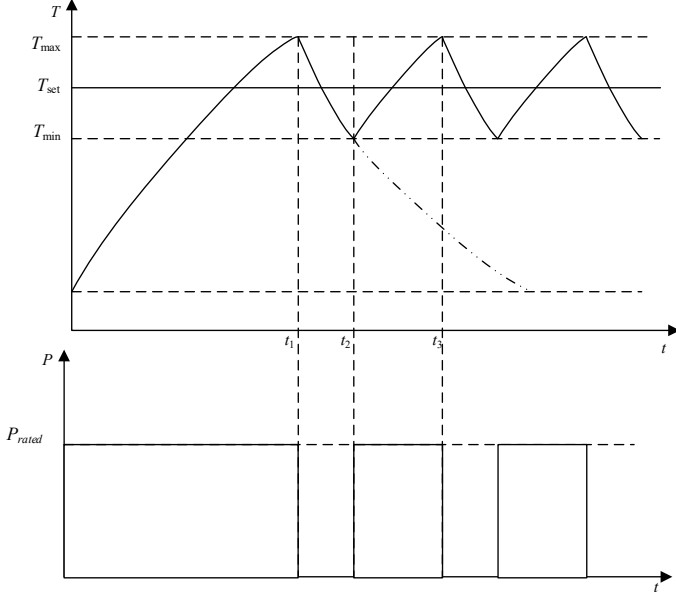

**Figure 5.** Variation curve of electric water heater temperature to power input.

### 3.2. Linearization

In order to simplify the calculation, we intend to linearize the temperature curve in Figure 5 and prove the rationality.

The process of temperature change is slow and the time for equipment to participate in demand response is relatively short, so it can be considered that the external temperature is constant. That is, $T_{out}{}^t = T_{out}{}^{t+1}$. The temperature range of the room(chamber) is set to $[T_{min}, T_{max}]$, the control cycle of $P_{ele}$ is $t_{cyc}$, and the time interval(i.e., time step) is $\Delta t$. The length of time of electric power input is $t_{on}$, and the length of time without electric power input is $t_{off}$. Substitute $T_{min}$, $T_{max}$ into Equation (10) for reception calculation, we get:

$$
\begin{aligned}
T_{max} &= (T_{out} + \eta R_1 P_{ele})\left(1 - e^{-\frac{t_{on}\Delta t}{R_1 C_e}}\right) + T_{min} e^{-\frac{t_{on}\Delta t}{R_1 C_e}} \\
T_{min} &= T_{out}\left(1 - e^{-\frac{t_{off}\Delta t}{R_1 C_e}}\right) + T_{max} e^{-\frac{t_{off}\Delta t}{R_1 C_e}} \\
t_{cyc} &= t_{on} + t_{off}
\end{aligned}
\tag{11}
$$

After solution, $t_{on}$ and $t_{off}$ are described by:

$$
\begin{aligned}
t_{off} &= \frac{R_1 C_e}{\Delta t} \ln\left(\frac{T_{max} - T_{out}}{T_{min} - T_{out}}\right) \\
t_{on} &= \frac{R_1 C_e}{\Delta t} \ln\left(\frac{T_{min} - T_{out} - \eta R_1 P_{ele}}{T_{max} - T_{out} - \eta R_1 P_{ele}}\right)
\end{aligned}
\tag{12}
$$

To describe the temperature change of each iteration by the ratio of the time step $\Delta t$ to $t_{on}$ and $t_{off}$, we obtain:

$$
\begin{cases}
T_{in}{}^{t+1} = T_{in}{}^t + \frac{\Delta t}{t_{on}}(T_{max} - T_{min}) & s = 1 \\
T_{in}{}^{t+1} = T_{in}{}^t - \frac{\Delta t}{t_{off}}(T_{max} - T_{min}) & s = 0
\end{cases}
\tag{13}
$$

### 3.3. Rationality of Linearization

Take the actual running condition of an electric water heater as an example. Assume that an electric water heater is heated from 30 °C to 50 °C, and after that the temperature is controlled from $T_{min}$ to $T_{max}$; the operation parameters of the electric water heater are as below: $t_{on}$ = 20 min, $t_{off}$ = 20 min, $P_{ele}$ = 2000 W, $T_{min}$ = 50 °C, $T_{max}$ = 60 °C, $T_{out}$ ≡ 20 °C, $T_{in}^0$ = 30 °C. It is available from (12) that: $R_1 = 3.5 \times 10^{-2}$ °C/W; $C_e = 1.192 \times 10^5$ J/°C. The shorter the communication time step, the more accurate the model and the more timely the control are. But at the same time, the communication pressure and the construction cost of AMI will increase. In this paper, the time step $\Delta t$ is 1 min.

Simulation is conducted on MATLAB R2016a (MathWorks, Natick, MA, USA) and the variation of temperature along time is shown in Figure 6. The solid line represents the results of the first-order ETP model, and the dotted line represents the results of the linearized ETP model.

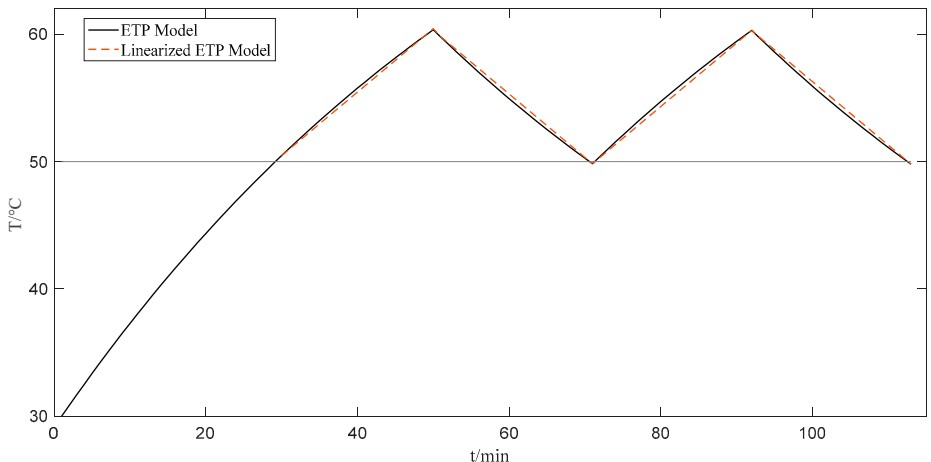

**Figure 6.** Comparison of temperature curves between linearized model and original ETP model.

Calculate the root mean square error (RMSE) of the two curves, and the smaller *RMSE is*, the less the influence of substitution will be.

$$RMSE(X, Y) = \sqrt{\frac{1}{N} \sum_{i=1}^{N} (X_i - Y_i)^2} \tag{14}$$

As the calculated RMSE between the linearized ETP model and the first-order ETP model within 113 min is only 0.2751 °C, it can be concluded that the linearized model fits well with the original model and will not bring significant change to the related results. Actually, the smaller the heating time $t_{on}$ is or the greater the cooling time $t_{off}$ is, the smaller RMSE is. Considering the actual situation of electric water heater, the duration of heating process of the equipment is generally much shorter than the duration of cooling process, that is, ton < toff.

## 4. Control Strategy of Electric Heating Loads Based on Virtual State of Charge (VSOC) Priority List

### 4.1. Proposal of the Control Strategy

Set the following assumptions:

The internal and external environment do not change when the control right is transferred; the energy conversion of electric heating equipment is 100%; the energy loss only comes from the difference between the chamber temperature and outside environment; the refresh time interval of communication data is $\Delta t$.

The following analysis still takes the electric water heater as an example. When a certain number of electric water heaters are controlled at the same point, the charge and discharge power of the *j*th one can be described by:

$$P^j_{\text{disc}}(t) = -S^j(t)P^j_{\text{rated}} + P_{\text{base}}{}^j$$
$$P^j_{\text{char}}(t) = S^j(t)P^j_{\text{rated}} - P_{\text{base}}{}^j \tag{15}$$

where the superscript *j* represents the *j*th VES.

At time *t*, the *j*th VES (VSOC$^j$) is like formula (8). After the model preprocessing of discretization and linearization, bring (13) into (7) to derive the formula (16):

$$\text{VSOC}^j(t) = \frac{Q(t)}{Q_{\text{capacity}}} = \frac{C^j(T^j(t) - T^j_{\text{min}})}{C^j(T^j_{\text{max}} - T^j_{\text{min}})} = \frac{T^j(t) - T^j_{\text{min}}}{T^j_{\text{max}} - T^j_{\text{min}}} \tag{16}$$

In (16), electrical parameters are described by thermal parameters, and interconversion from electrical parameters to thermal parameters is completed in time domain. The relationship between linearized temperature curve, VSOC curve and electric power is shown in Figure 7.

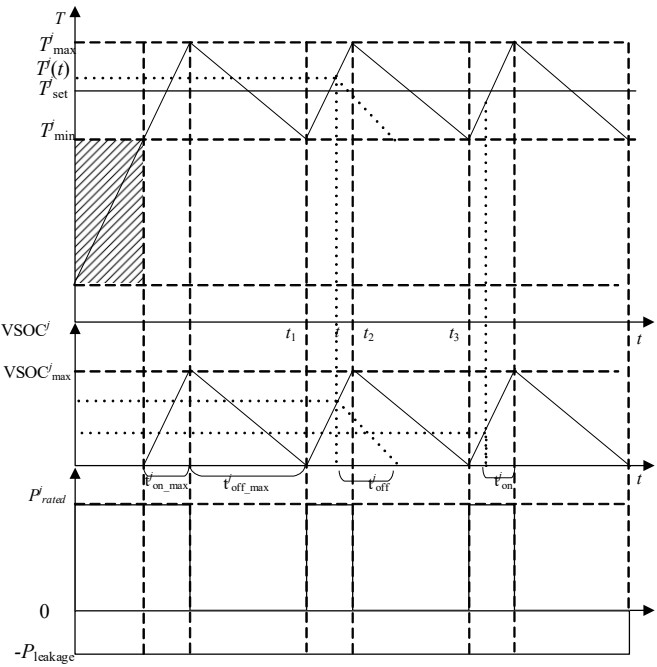

**Figure 7.** The relationship between linearized temperature curve, virtual state of charge (VSOC) curve and electric power.

The shaded portion indicates that the temperature of VES equipment has not reached the protocol value. $T^j(t)$ represents the temperature of *j*th VES at time *t*; $t^j_{\text{off}}$ and $t^j_{\text{\_on}}$ are remaining discharge and charge time of *j*th VES at time *t*; $t^j_{\text{off\_max}}$ and $t^j_{\text{on\_max}}$ are the maximum discharge and charge time of *j*th VES.

The recursion formula of VSOC$^j$ is as follows:

$$\text{VSOC}^j(t+1) - \text{VSOC}^j(t) = \frac{T^j(t+1) - T^j_{\text{min}}}{T^j_{\text{max}} - T^j_{\text{min}}} - \frac{T^j(t) - T^j_{\text{min}}}{T^j_{\text{max}} - T^j_{\text{min}}} \tag{17}$$

By substituting (13) into (17), it can be concluded that VSOC$^j$ is related to time step:

$$\text{VSOC}^j(t+1) = \frac{\Delta t}{S^j t^j_{\text{on\_max}} - (1 - S^j)t^j_{\text{off\_max}}} + \text{VSOC}^j(t) \tag{18}$$

Charge/discharge time are important constraint indexes. On the one hand, they can ensure that the VES will not charge or discharge excessively, which means that the temperature fluctuation of the electric water heater is within the set range. On the other hand, they can ensure that the state of the device will not change during the time step, which shuns affecting the control accuracy and bringing unnecessary grid side fluctuations. According to Figure 7, we can obtain:

$$
\begin{aligned}
\frac{t^j_{\text{off}}}{t^j_{\text{off\_max}}} &= \frac{\text{VSOC}^j(t) - \text{VSOC}^j_{\text{min}}}{\text{VSOC}^j_{\text{max}} - \text{VSOC}^j_{\text{min}}} \\
\frac{1 - t^j_{\text{on}}}{t^j_{\text{on\_max}}} &= \frac{\text{VSOC}^j(t) - \text{VSOC}^j_{\text{min}}}{\text{VSOC}^j_{\text{max}} - \text{VSOC}^j_{\text{min}}}
\end{aligned}
\tag{19}
$$

$$
\begin{aligned}
t^j_{\text{off}} &= \frac{T^j(t) - T^j_{\text{min}}}{T^j_{\text{max}} - T^j_{\text{min}}} t^j_{\text{off\_max}} = \text{VSOC}^j(t) t^j_{\text{off\_max}} \\
t^j_{\text{on}} &= \frac{T^j_{\text{max}} - T^j(t)}{T^j_{\text{max}} - T^j_{\text{min}}} t^j_{\text{on\_max}} = (1 - \text{VSOC}^j(t)) t^j_{\text{on\_max}}
\end{aligned}
\tag{20}
$$

To sum up, taking discharge as an example, the control strategy of VES are:

$$
\begin{cases}
P^j_{\text{disc}}(t) = -S^j(t) P^j_{\text{rated}} + P_{\text{base}}{}^j \\
\text{VSOC}^j(t+1) = \dfrac{\Delta t}{S^j t^j_{\text{on\_max}} - (1 - S^j) t^j_{\text{off\_max}}} + \text{VSOC}^j(t) \\
t^j_{\text{off}} = \dfrac{T^j(t) - T^j_{\text{min}}}{T^j_{\text{max}} - T^j_{\text{min}}} t^j_{\text{off\_max}} = \text{VSOC}^j(t) t^j_{\text{off\_max}}
\end{cases}
\tag{21}
$$

Taking the demand side response of discharge condition as an example, the principle of the control strategy for VESs is to control the switching state mainly based on the sequence of VSOC values, that is, the unit with higher VSOC is shut down preferentially. The main objective function is to meet the power shortage in each time step. The marginal limit conditions are that the discharge time is longer than the time step and VES do not overcharge or overdischarge.

The specific control function at time *t* are

$$
\begin{cases}
P^t_{\text{s}} \leq \min\limits_{Q} \sum P^t_{\text{disc}}(j_n) \\
0 \leq \text{VSOC}^t(j_n) \leq 1 \\
t^j_{\text{off}} \geq \Delta t
\end{cases}
\tag{22}
$$

where $P^t_{\text{s}}$ represents the power shortage at time *t*; Q represents the set arranged from large to small according to VSOC; $j_n$ is an element of set Q, and *n* represents the order of *j* in the new set.

### 4.2. Simulation of the Strategy

MATLAB is used as the simulation platform to verify the control effect of trunked dispatching of the electric heating loads with different parameters and working states. The program mainly includes the following steps: data refreshment, dealing with VSOC off-limit problem, generating control queue Q based on VSOC values, calculating whether the power shortage is satisfied, handling the switch state of controlled energy storage and updating the states of VES iteratively. The parameters of the example are: the amount of electric water heaters under control is 100; the initial value of VSOC is uniformly distributed from 0 to 1; the switching function is 0~1 integer distributed; the time step is 1min; the rated power of the equipment is uniformly distributed from 1.5 to 2.5 kW. For each water heater, the maximum charge and discharge time are 15~25 min and 30~50 min uniformly distributed, respectively; the protocol minimum and maximum temperature are 45~55 °C and 55~65 °C uniformly distributed, respectively.

Suppose the power shortage in the power system is 30 kW and the protocol control time is 30 min, and the control result is shown in Figure 8. After analysis, it can be found that the response power can

satisfy the power shortage well in a short time, but there is an excessive response at 23 min, as shown in Figure 8a. At the same time, a large number of VSOC values reach the limit in Figure 8b, which shows that there is a certain relationship between the excessive response of power and a large number of VESs reaching the limits simultaneously.

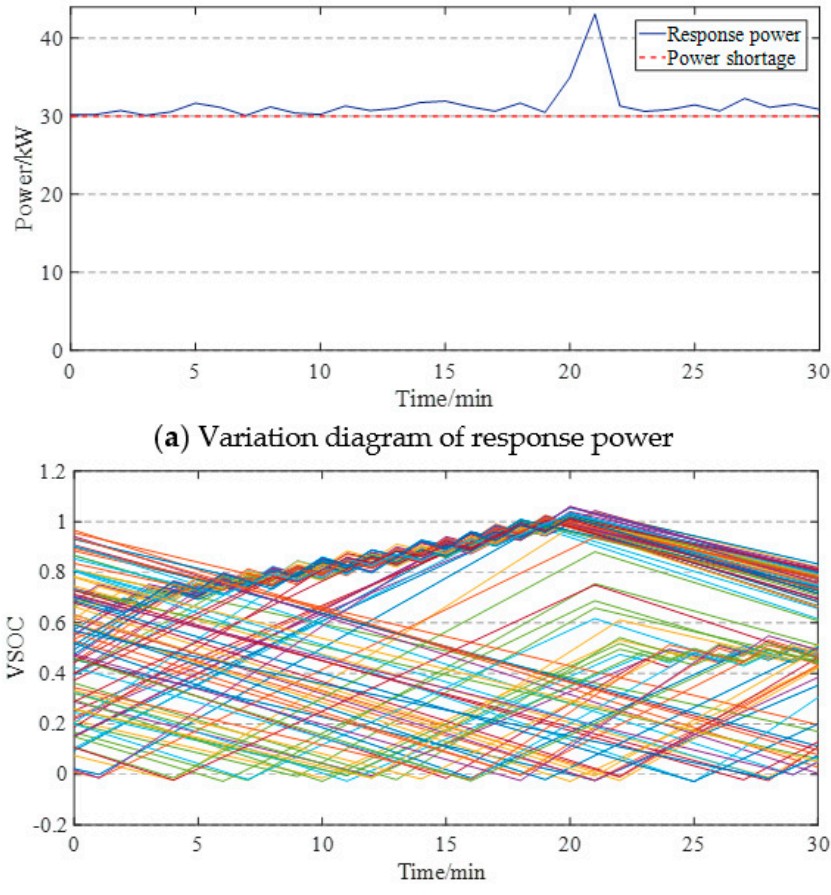

**(a)** Variation diagram of response power

**(b)** Variation diagram of VSOC of 100 electric water heaters

**Figure 8.** The response variation of power and VSOC in 30 min.

Suppose the power shortage is 30 kW and the protocol control time is 180 min. The control result is shown in Figure 9. It is found that the response power basically satisfies the power shortage in a long time, but at the same time there are more excessive responses and insufficient responses. By comparing Figure 9a,b, we come to the same conclusion as Figure 8: when a large number of VSOC values are concentrated and near the limit value, they will lead to excessive or insufficient response. Especially starting at 135 min, since most VESs are close to the limit of VSOC = 0, and in order to maintain the marginal condition VSOC > 0, a large number of VESs are forced to open and charge, resulting in insufficient response over a period of time.

Above all, the control strategy based on the VSOC priority list can achieve a relatively stable power response in a short time. However, limited by the state of VES, the control results over a long period of time are yet to be adjusted and optimized.

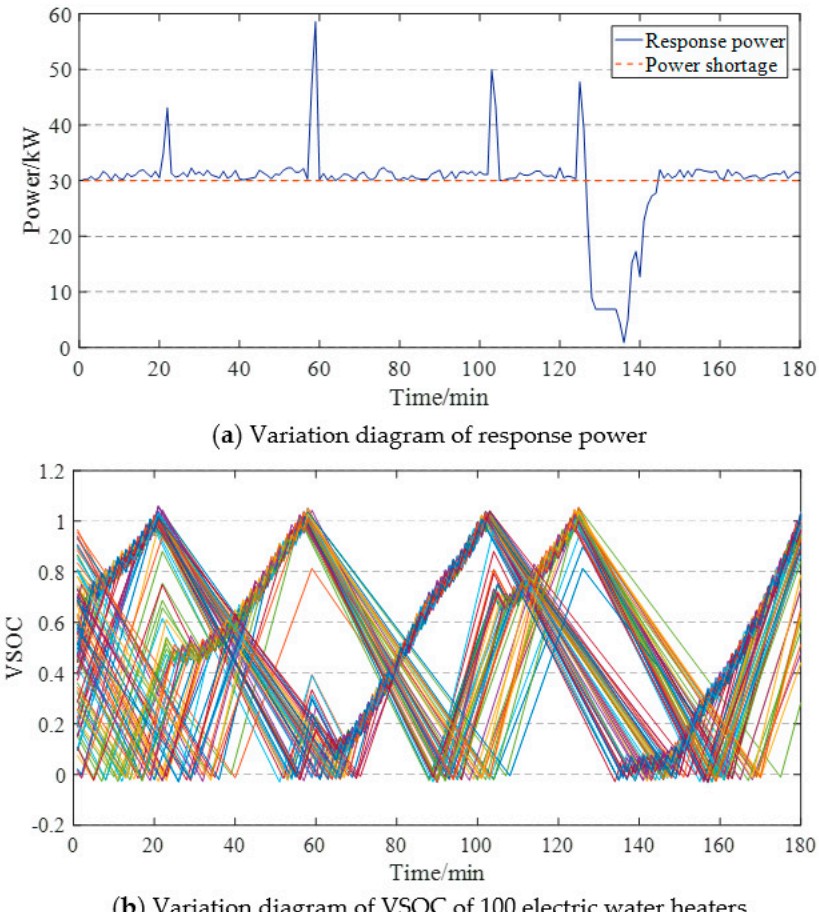

**Figure 9.** The response variation of power and VSOC in 180 min.

*4.3. Discussion*

After observing Figures 8 and 9, it would be found that when the power response is excessive, more electric water heaters are forced to close and their equivalent VESs discharge in advance because their VSOC values reach the upper limit, which leads to load peak of excessive response. When the response is inadequate, more electric water heaters are in uncontrolled state, namely, the relevant equipment is closed and there is inadequate load capacity for respond. At the same time, we found that before insufficient response of VES, there is always a large excessive response. The excessive response power is so large that the state of charge and discharge is changed in advance, causing insufficient discharge capacity and insufficient response.

By analyzing Figure 8 and the first 30 min of Figure 9, it is found that the control effect is better when the control right is transferred for a short time than that for a long period of time. By comparison, it is found that the states of the VESs are more dispersed in a short period of time, while more concentrated after long-term control.

As shown below, the control effect is affected by the diversity of VES.

The degree of distribution of virtual state of charge of virtual energy storage is defined. It is expressed by the standard deviation of VSOC. It reflects the diversity of virtual VES. The greater the standard deviation, the higher diversity of related VES.

By applying the relevant parameters in 4.2, the variation of VES diversity in 180 min can be obtained, as shown in Figure 10. It is obvious that with the increasing of transfer time of control right, the diversity of VES converges to a smaller value oscillatorily. The histograms of VES distribution at special time points 10 min, 55 min, 95 min and 115 min are shown in Figure 11. At 10 min, the distribution of VES is relatively uniform, but at 55 min, 95 min, and 115 min, it is relatively

concentrated and the diversity is lower. Compared with Figure 9, 55 min, 95 min, and 115 min are the time points when excessive or insufficient response occurs.

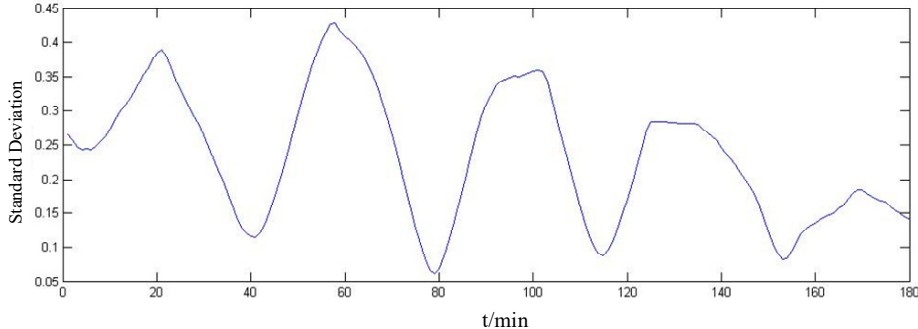

**Figure 10.** The variation of VES diversity.

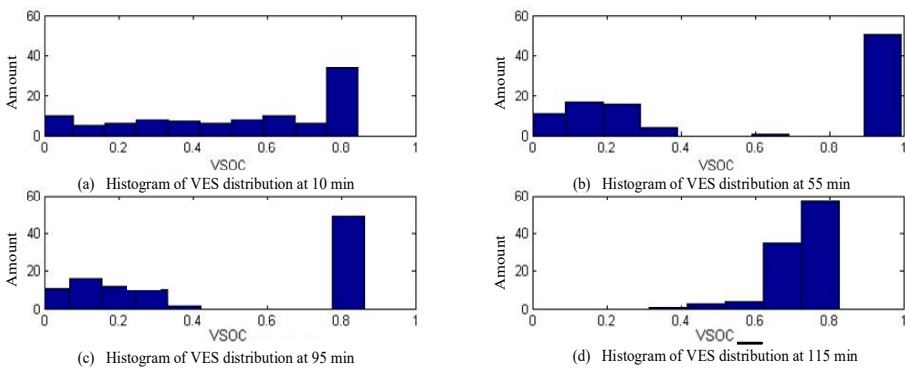

**Figure 11.** Histograms of VES distribution at different time points.

With increasing control time, VES diversity of electric water heaters is decreased, and that is before excessive response or insufficient response is low. Obviously, the diversity state of VES leads to excessive and insufficient response in the transfer process of control right.

In summary, in the response process, the above VSOC-priority VES control strategy makes the state of VESs convergent and thus reduces its diversity, which leads to worse results than expected, namely, excessive or insufficient response occurs frequently after long-term control right transfer.

## 5. Optimized Control Strategy of Electric Heating Loads Based on VSOC Priority List

### 5.1. Proposal of the Optimized Control Strategy

Based on the above analysis, the diversity of VES states directly affects the control results.

In (22), the switching state of set Q is refreshed within each communication time step in the original control strategy, which will inevitably lead to the discharge of VES with higher VSOC and charge of VES with lower VSOC，making the states of VESs synchronised, resulting in the reduction of the diversity of VESs and directly affecting the control effect.

In order to make the charge and discharge of each VES more complete, the improved control strategy is to reduce the switching times of VES, trying to change the active control to passive control according to limit value. To this end, the improved restrictive conditions are:

$$\begin{cases} P^t{}_s \leq \min\sum_{A} P^t{}_{disc}(j_n) \\ P^t{}_s - \sum_{A} P^t{}_{disc}(j_n) \leq \min_{\complement_Q A} \sum P^t{}_{disc}(j_n) \\ 0 \leq \text{VSOC}^t(j_n) \leq 1 \\ t^j{}_{off} \geq \Delta t \end{cases} \quad (23)$$

where A represents the set of controlled VESs arranged from large to small according to VSOC; $C_QA$ represents the complement set of A with Q as the complete set. The new restriction condition indicates that the VES discharging during the last time step is preferentially controlled to continue discharging, and then the power shortage is supplemented based on the order of VSOC values.

### 5.2. Simulations of the Optimized Strategy

The example parameters in Section 4 and the optimized control strategy based on VSOC priority list for electric water heaters are used to draw the VES diversity variation curve as shown in Figure 12.

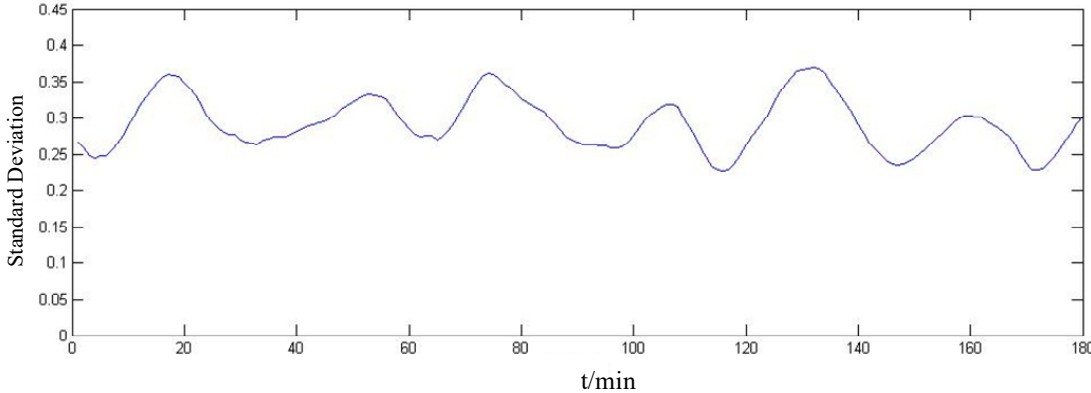

**Figure 12.** Variation of VES diversity of electric water heaters (after optimized).

Through quantitative calculation and comparison with Figure 10, it can be found that the diversity of VES obtained by the optimized control strategy is maintained well, the standard deviation oscillates within a stable range, and the amplitude is much smaller than that of the original control strategy. The histograms of VES distribution at 45 min, 90 min, 135 min and 180 min are shown in Figure 13. Obviously, the distribution of VES is more uniform and the diversity is better.

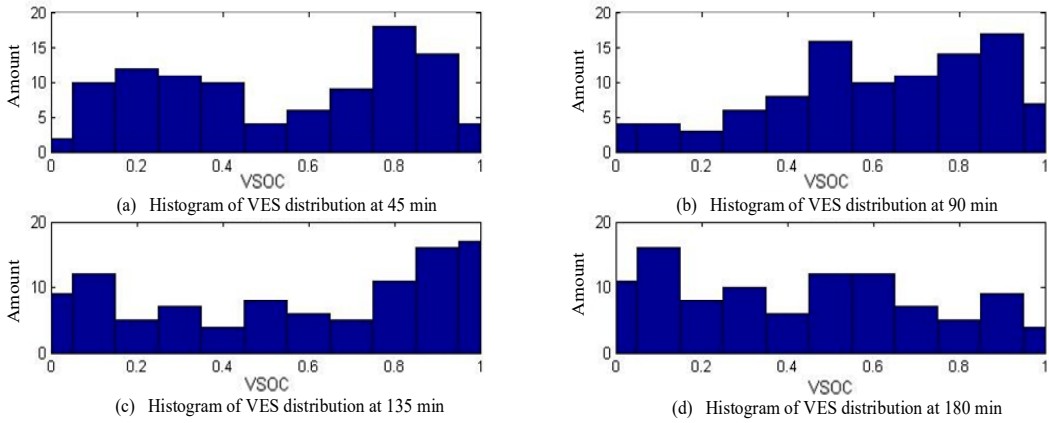

(a) Histogram of VES distribution at 45 min

(b) Histogram of VES distribution at 90 min

(c) Histogram of VES distribution at 135 min

(d) Histogram of VES distribution at 180 min

**Figure 13.** Histograms of VES distribution at different time points (after optimized).

The control effect is shown in Figure 14. The simulation result shows that the optimized control strategy has better control effect because long-term transfer of control rights does not result in insufficient or excessive response. The VSOC values of VESs are relatively uniform in the whole process, and there is no convergence of the states of VES. The power shortage can be reduced steadily by using the optimized control strategy to control electric heating loads.

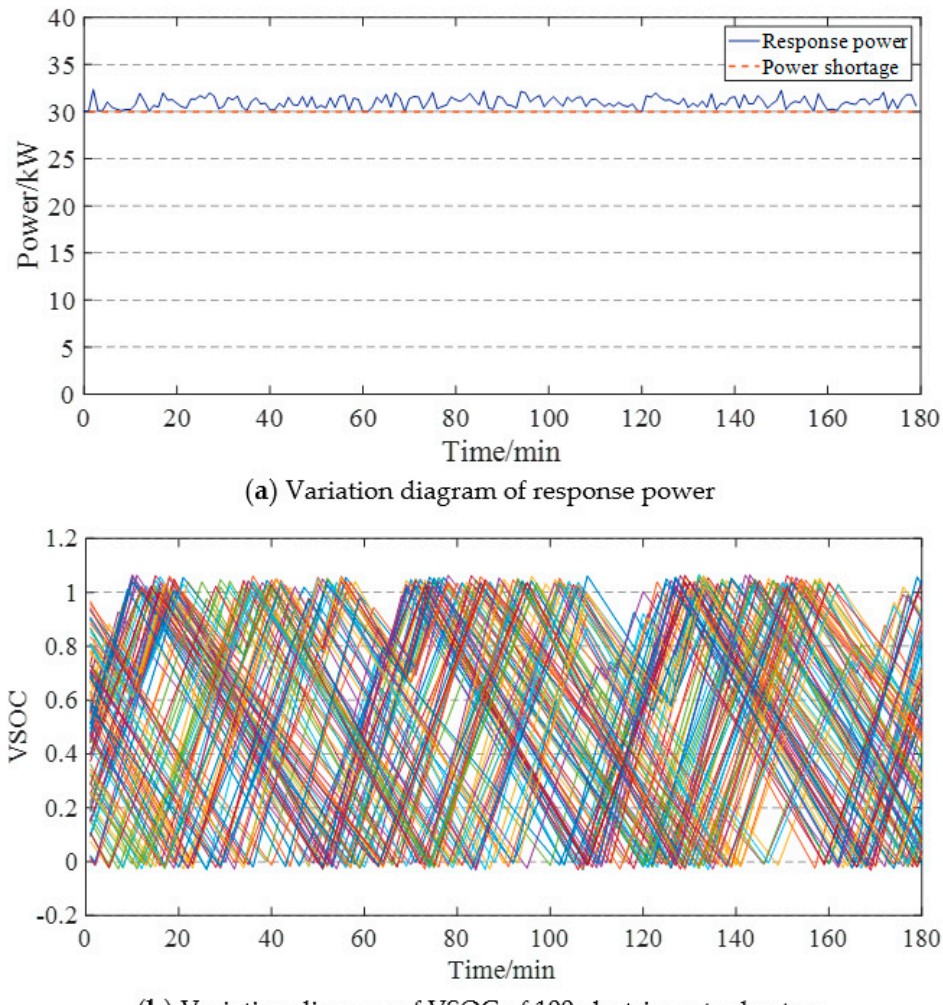

**(a)** Variation diagram of response power

**(b)** Variation diagram of VSOC of 100 electric water heaters

**Figure 14.** Control effect of optimized control strategy for electric water heaters.

The control effect of the optimized control strategy based on a VSOC priority list for electric water heaters under fluctuating power shortage is shown in Figure 15. As can be seen from the figure, the response power still tracks power shortage well, which shows that the control strategy proposed in this paper is also applicable to power grid with power shortage fluctuations.

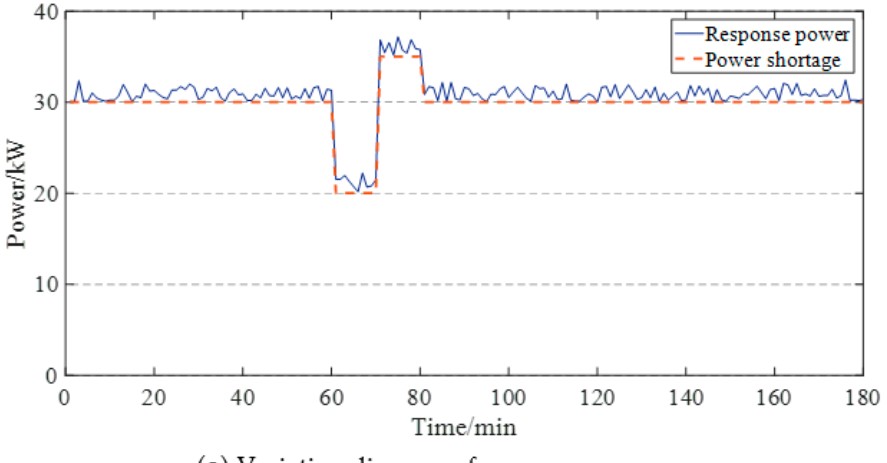

**(a)** Variation diagram of response power

**Figure 15.** *Cont.*

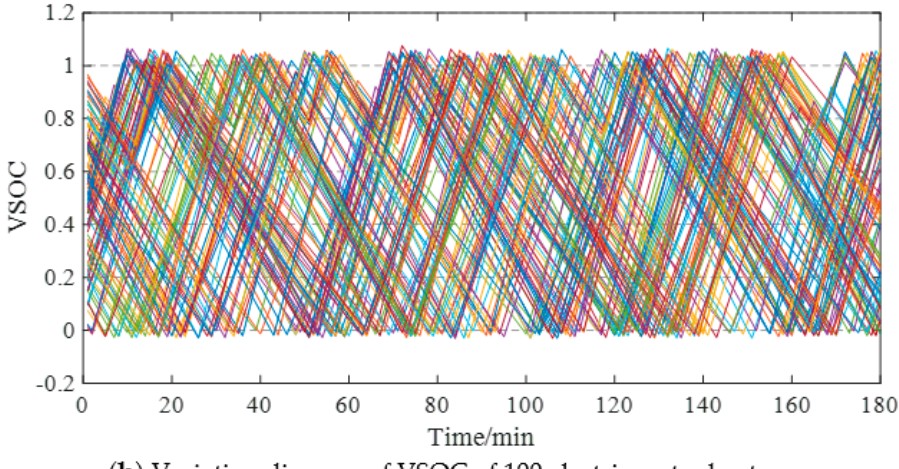

(**b**) Variation diagram of VSOC of 100 electric water heaters

**Figure 15.** Control effect of optimized control strategy under fluctuating power shortage.

## 6. Conclusions

In this paper, we establish a complete VES index system and propose a VES model which can reflect the practical electro-thermal exchange. The model is mainly divided into two parts: electrical parameters and thermal parameters, reflecting the impact of electric heating loads scheduling variation on the distribution network. The model is discretized to reduce communications traffic and linearized to simplify calculation. Taking the electric water heater as an example, a control strategy based on VSOC priority list is proposed, and simulation results show that this method can reduce the power shortage of the grid to a certain extent. By analyzing the insufficiency of the strategy, the optimized VSOC priority list control strategy, which optimizes the control effect of long-time scheduling, is put forward. The optimized control strategy can maintain the diversity of VES well and make it possible for the electric water heater to track the power shortage of the grid for a long time. Simulation examples are designed to verify the superiority and effectiveness of the proposed optimized control strategy.

The control strategy proposed in this paper is not only applicable to the electric water heater, but also to other electric heating loads that maintains the temperature of the room (chamber) in a specific temperature range by controlling the on-off mode of the resistance wire. For other types of electric heating loads, it is only necessary to modify the electrical parameters of the VES model and propose its control strategy in a targeted manner. The research in this paper is helpful to build a multi-VES system which can participate in the trunked dispatching of the power grid and promote the development of demand side response. The cost, benefit and pricing mechanism of demand response can be analyzed in the following studies.

**Author Contributions:** Conceptualization, Y.C.; Data curation, S.X.; Formal analysis, S.X.; Investigation, Y.Z.; Methodology, Y.C.; Project administration, Y.C.; Software, S.X.; Supervision, Y.Z.; Writing—original draft, Y.C., S.X.; Writing—review and editing, W.H. and R.Z.

**Funding:** This work is supported by SGCC program: Research on Extensive Application and Benefit Evaluation of Typical Power Substitution Technology Considering Power Quality Influence (52182018000H) and Lanzhou Jiaotong University-Tianjin University Innovation Fund Program: Research on Management Strategies of Power Quality Problems from Thermostatically Controlled Appliances (2018058).

**Conflicts of Interest:** The authors declare no conflicts of interest.

## Abbreviation

| | |
|---|---|
| $P_{ele}$ | electric power |
| $\eta$ | refrigeration or heating efficiency |
| $T_{out}$ | ambient temperature |
| $T_{in}$ | room (chamber) temperature |
| $C_e$ | heat capacity of medium |
| $R_1$ | heat transfer resistance of energy between the interior and the exterior environment of the room (chamber) |
| $P_{heat}$ | VES power supply |
| $P_{leakage}$ | leakage current |
| $P_{base}$ | electric power before control right transfering |
| $P_{disc}$ | discharging power |
| $P_{char}$ | charging power |
| $S(t)$ | switching state at time $t$ |
| $t_{on}$ | charge time |
| $t_{off}$ | discharge time |
| $Q_{capacity}$ | maximum capacitance of VES |
| $T_{max}$ | protocol maximum temperature |
| $T_{min}$ | protocol minimum temperature |
| $Q(t)$ | charge capacity at time $t$ |
| $P_{heat1}$ | heating power |
| $S(t-1)$ | switching state at the last time step |
| $P_{rated}$ | rated electric power of electric water heater |
| $\Delta t$ | time step |
| $T_{in}^{t+1}$ | internal temperature of the room(chamber) at time $t + 1$ |
| $RMSE$ | root mean square error of two curves, |
| $(Symbol)^j$ | (Symbol) of $j$th VES |
| $t^j_{off}$ | remaining discharge time of $j$th VES at time $t$ |
| $t^j_{h\_on}$ | remaining charge time of $j$th VES at time $t$ |
| $t^j_{off\_max}$ | maximum discharge time of $j$th VES |
| $t^j_{on\_max}$ | maximum charge time of $j$th VES |
| $P^t_s$ | power shortage at time $t$ |
| Q | the set arranged from large to small according to VSOC |
| $j_n$ | the order of $j$ in the new set |
| $C_Q A$ | the complement set of A with Q as the complete set |

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
