# Peer review of "Control Strategy of Electric Heating Loads for Reducing Power Shortage in Power Grid"

_processes, doi:10.3390/pr7050273_

Reviewer 1 Report

This paper proposes a virtual energy storage model suitable for electric heating loads with different electrical and thermal parameters. In particular, the model is processed by discretization and linearization. and a control strategy of electric heating load based on virtual state of charge priority is used. The simulation results show that the proposed method can reduce power storage in power grid over a long period of time and realize power response of the power grid.

As a general comment, I think the paper makes an interesting contribution to the literature by suggesting this new strategy. At the same time, I think the paper needs some minor improvements before to be published in this journal.

Broad comments

1)      In the introduction, I would suggest the authors to modify the sentences that start with the reference number.

2)      In order to let the paper more readable, I would suggest the authors to add a table with the meaning of the main symbols used.

3)      In section 3.3, could the authors explain how they choose the parameters values? In particular, how is chosen Delta t? Moreover, do the results change if Delta t changes? Please explain.

4)      I would suggest the authors to add a section were the proposed strategy is compared to other strategies underlining the advantages and disadvantages. Many algorithms are used in literature as listed in

a.       Gallo, G., Ponta, L., Cincotti, S, “Profit-based O&M strategies for wind power plants”, (2012), 6254699, 9th International Conference on the European Energy Market, EEM 12

b.       Miranda, V., Srinivasan, D., Proença, L.M., “Evolutionary computation in power systems”, (1998), International Journal of Electrical Power and Energy Systems, 20 (2), pp. 89-98.

Specific comments

In the figures the labels size should be increased in order to let the figures more readable.

In the figures 8a, 9a, 14a, I would suggest the authors to use not only different colour but also different line style or marker so that also in black and white the figures are readable.

Author Response

Response to reviewer 1

(Manuscript ID: processes-476066)

The authors gratefully thank you for the comments and suggestions. Those comments are all valuable and very helpful for revising and improving our paper, as well as the important guiding significance to our researches. We have responded to all the comments as follows.

Comment 1:

In the introduction, I would suggest the authors to modify the sentences that start with the reference number.

Response 1:

Thank you very much for your comments. The sentences that start with the reference number have been modified

Comment 2:

In order to let the paper more readable, I would suggest the authors to add a table with the meaning of the main symbols used.

Response 2:

Thank you very much for your comments. A table introducing the meaning of main symbols used in this paper is added in Appendix (line 437)

Comment 3:

In section 3.3, could the authors explain how they choose the parameters values? In particular, how is chosen Delta t? Moreover, do the results change if Delta t changes? Please explain.

Response 3:

Thank you very much for your comments. The simulation parameters ton, toff, Tmin, Tmax, Tout are obtained according to the actual situation in life. Different equipments and environments will correspond to different parameters. Any parameters reasonable is Ok. And R1, Ce can then be calculated.

What’s more, the parameter delta t have been analyzed in line 225-227.

Comment 4:

I would suggest the authors to add a section were the proposed strategy is compared to other strategies underlining the advantages and disadvantages.

Response 4:

Thank you very much for your suggestion. The advantages and disadvantages of methods in other studies and the research status have been introduced in more detail, and the contribution of this paper have been further emphasized in the introduction.

Specific comments

In the figures the labels size should be increased in order to let the figures more readable.

In the figures 8a, 9a, 14a, I would suggest the authors to use not only different colour but also different line style or marker so that also in black and white the figures are readable.

Response:

Thank you very much for your suggestion. The simulation result graphs are redrawn to be seen more clearly

In the figures 8a, 9a, 14a, the line style of power shortage is changed.

Special thanks to you for your good comments.

Reviewer 2 Report

Review: Control Strategy of Electric Heating Loads for Reducing Power Shortage in Power Grid

The paper presents a modelling and simulation study of control strategies for the use of thermostatically controlled loads (TCL) in demand response. The main contribution is the development of a control strategy. This topic is in the scope of the journal Processes.

While the idea of the manuscript is relevant and timely, the current presentation makes it very difficult to assess the merit of the work. Thus, I have to recommend a substantial rewrite and resubmission.

Here is a list of some suggestions which could be part of an improved manuscript for potential resubmission in the future:

1. A number of sentences are difficult to understand due to the use of unusual words and phrases. In addition, the grammar and spelling of the manuscript need to be improved. It might be worthwhile to have the paper checked by a native English speaker.

2. All abbreviations should be explained in the text, e.g. ETP in line 43.

3. Some expressions are not very scientific and need to be clarified. For example, the phrase 'a long time' in line 373 should be clarified and put into relation to the number of water heaters and the average grid power. In addition, 'slowly' in line 119 should be clarified.

4. The 'virtual power storage device'  in line 37-38 should be explained with the comfort temperature range and that this range is used to change the setting of the TCLs.

5. From the introduction it is not clear what the research gap is that is addressed in this paper. Other researchers have used TCLs for demand side response. Please clarify what research gap is filled.

6. In lines 68ff, it should be made clearer what is presented in this paper, i.e. a control strategy for a large cluster of TCLs.

7. In lines 79ff, the operation of the VES should be more clearly explained. In addition, it is not clear which operation is charging and which is discharging. Is the system fully charged or discharged when the minimum temperature is reached?

8. Explain the physical processes which are involved in the electric heating load and how they are modelled in Figure 1. Are you modelling a room or a hot water heater with storage? What would be the difference? Are you including occupancy or other effects on temperature? Why or why not?

10. All terms in equations should be defined, e.g. C_e in Eq. 1, T_min and T_max in Eq. 5 and P_leakage aren't defined.

11. The role of P_base is unclear and should be clarified.

12. Explain the role of S(t-1) in Eq. 8.

13. Why do you need to simplify the calculation? The model is not that complicated.

14. In line 187, Figure 4 should be Figure 5.

15. In line 211, you should describe the system for which the values are given.

16. In line 222, why is t_on<t_off? Explain.

17. In line 232, you haven't mentioned multiple TCL before. This makes this sentence very hard to understand. You probably should explain the intended use at the end of the introduction.

18. The control algorithms 21 and 22 need to be better explained. In particular, the difference between them is not sufficiently clear.

19. What is the difference between the results in Figures 8 and 9? According to the text both are for a shortage of 30kW.

20. Explain why in Figure 8a the response power is always higher than the power shortage.

21. It is not clear why the VSOC in Figures 8b and 9b is not decreasing for all units. How can some of the units charge if there is a power shortage?

22. Why are the TCLs synchronising? It might be interesting to evaluate the response to a fluctuating power shortage.

23. Can you quantify the effect of the demand response?

Author Response

Response to reviewer 2

(Manuscript ID: processes-476066)

The authors gratefully thank you for the comments and suggestions. Those comments are all valuable and very helpful for revising and improving our paper, as well as the important guiding significance to our researches. We have responded to all the comments as follows. The main changes made in the manuscript are highlighted with red.

Comment 1:

A number of sentences are difficult to understand due to the use of unusual words and phrases. In addition, the grammar and spelling of the manuscript need to be improved. It might be worthwhile to have the paper checked by a native English speaker.

Response 1:

Thank you very much for your comments. Some grammatical problems in the paper have been revised. If there are still grammar and spelling problems, we will continue to revise it.

Comment 2:

All abbreviations should be explained in the text, e.g. ETP in line 43.

Response 2:

We are very sorry for our negligence. The abbreviations have been explained in the text. ETP have been explained in line 46.

Comment 3:

Some expressions are not very scientific and need to be clarified. For example, the phrase 'a long time' in line 373 should be clarified and put into relation to the number of water heaters and the average grid power. In addition, 'slowly' in line 119 should be clarified.

Response 3:

Thank you very much for your comments. The expressions now have been clarified. The phrase 'a long time' has been clarified in line 73-76, and the word 'slowly' have been replaced by ‘smoothly‘ in line 131.

Comment 4:

The 'virtual power storage device' in line 37-38 should be explained with the comfort temperature range and that this range is used to change the setting of the TCLs.

Response 4:

Thank you very much for your suggestion. The 'virtual power storage device' have been explained in line 38-40.

Comment 5:

From the introduction it is not clear what the research gap is that is addressed in this paper. Other researchers have used TCLs for demand side response. Please clarify what research gap is filled.

Response 5:

Thank you very much for your comments. The advantages and disadvantages of methods in other studies and the research status have been introduced in more details, and the contribution of this paper has been further emphasized in the introduction.

Comment 6:

In lines 68ff, it should be made clearer what is presented in this paper, i.e. a control strategy for a large cluster of TCLs.

Response 6:

Thank you very much for your suggestion. The control strategy of trunked dispatching of electric heating loads has been clarified clearer in line 77-79

Comment 7:

In lines 79ff, the operation of the VES should be more clearly explained. In addition, it is not clear which operation is charging and which is discharging. Is the system fully charged or discharged when the minimum temperature is reached?

Response 7:

Thank you very much for your comments. The operation of the VES has been further explained in line 90-93. The state charging or discharging depends on the working states of the equipment before and after the control right transferring, as shown in equation (3). The decrease of VSOC does not necessarily means discharging. A equipment off after the control right transferring while off before control right transferring is not discharging.

In this paper, the concept 'fully charged' and 'fully discharged' are not proposed. However, this paper proposes a parameter VSOC with upper and lower limits. When the VSOC of the equipment reaches the lower limit, its working state will be transferred to the open state.

Comment 8:

Explain the physical processes which are involved in the electric heating load and how they are modelled in Figure 1. Are you modelling a room or a hot water heater with storage? What would be the difference? Are you including occupancy or other effects on temperature? Why or why not?

Response 8:

Thank you very much for your comments. The physical processes of second-order ETP model have been introduced in line 117-119. The model can be applied to both a room with electric heating loads and an electric water heater. Their difference is that the parameters Phear, R1, Ce, Tin, Tout, ton, toff, are different.

Occupancy or other effects on temperature are not included in this paper. Because considering other parameters will greatly increase the difficulty of modeling and the control. although it can increase the accuracy of the model. What’s more, the behavior of users is difficult to predict

Comment 10:

All terms in equations should be defined, e.g. C_e in Eq. 1, T_min and T_max in Eq. 5 and P_leakage aren't defined.

Response 10:

We are very sorry for our negligence. All terms in equations have been defined. Ce is defined in line 123. Tmin and Tmax are defined in line 158. Pleakage is defined in line 141. And a table introducing the meaning of main symbols used in this paper is added in Appendix (line 437).

Comment 11:

The role of P_base is unclear and should be clarified.

Response 11:

Thank you very much for your comments. The electric power before control right transfering is set as base power Pbase. An electric heating equipment not in use does not have discharge capability, and Pbase is 0.(line 144-145)

Comment 12:

Explain the role of S(t-1) in Eq. 8.

Response 12:

Thank you very much for your suggestion. S(t-1) represents the switching state at the last time step, 1 for running and 0 for stop (added in line 180-181).

Comment 13:

Why do you need to simplify the calculation? The model is not that complicated.

Response 13:

The model is linearized to simplify the calculation because:

(1) Using the simplified model does simplifies the calculation both in theoretical research and practical application.

(2) The error between the simplified model and the actual model is small.

(3) Using the simplified model can also reduce communications traffic of AMI.(added in line 193-194)

Comment 14:

In line 187, Figure 4 should be Figure 5.

Response 14:

We are very sorry for our negligence. It has been corrected in line 201.

Comment 15:

In line 211, you should describe the system for which the values are given.

Response 15:

Thank you very much for your suggestion. The system was described in line 203-207 and has been further described in line 221-222,.

Comment 16:

In line 222, why is t_on<t_off? Explain.

Response 16:

Thank you very much for your comments. ton is less than toff because the duration of heating process of electric water heater is generally much shorter than the duration of cooling process according to the actual situation. That is, ton < toff. (added in line 238-240)

Comment 17:

In line 232, you haven't mentioned multiple TCL before. This makes this sentence very hard to understand. You probably should explain the intended use at the end of the introduction.

Response 17:

Thank you very much for your suggestion. The intended use of the control strategy has been introduced in line 40-43 and line 77.

Comment 18:

The control algorithms 21 and 22 need to be better explained. In particular, the difference between them is not sufficiently clear.

Response 18:

Thank you very much for your comments. The optimized control algorithms 23 (22 in the formal version) have been further explained in line 367-369.

Comment 19:

What is the difference between the results in Figures 8 and 9? According to the text both are for a shortage of 30kW.

Response 19:

The simulation parameters in Figures 8 and 9 are the same, but the control times are different. Figure 8 can be considered as the 0-30 minute section taken from Figure 9.

Comment 20:

Explain why in Figure 8a the response power is always higher than the power shortage.

Response 20:

The electric power of each unit is a discrete variable with value Pele or 0. Therefore, the response power cannot be exactly equal to the power shortage. But it can just beyond the power shortage.

Comment 21:

It is not clear why the VSOC in Figures 8b and 9b is not decreasing for all units. How can some of the units charge if there is a power shortage?

Response 21:

Thank you very much for your comments. The main objective function is to meet the power shortage in each time step. A device may charge or discharge after control right transferring, but the total electric power have to be decreased. And a device whose VSOC reaches the lower limit have to be opened and charge.

Comment 22:

Why are the TCLs synchronising? It might be interesting to evaluate the response to a fluctuating power shortage.

Response 22:

Thank you very much for your comments. The reason TCLs synchronizing has been explained in line 359-362. A simulation of the optimized control strategy for electric heating loads responding to a fluctuating power shortage is analyzed in line 394-403.

Comment 23:

Can you quantify the effect of the demand response?

Response 23:

Thank you very much for your suggestion. The effect of demand response belongs to the category of economic study of power grid, which includes system benefit such as reducing peak load, maintaining power supply reliability and reducing investment in power plant construction and social benefit such as reducing energy consumption and reducing emissions of polluted gases.

This paper proposes a VES model suitable for electric heating loads with different electrical and thermal parameters and a trunked dispatching strategy based on the model and VSOC priority. This paper belongs to the category of technical research on demand response, so the economic research of demand response is not covered. However, in the conclusion section of this article we prospects the study of the cost, benefit and pricing mechanism of demand research according to the control strategy proposed in this paper.

Special thanks to you for your good comments.

Reviewer 3 Report

In my opinion, this paper presents useful and interesting results and it can be accepted for publication as long as the authors read it several times and correct typos/grammatical errors

Author Response

Some grammatical errors have been corrected. The authors gratefully thank you for your recognition of our work.